# NaF-PET Imaging of Atherosclerosis Burden

**DOI:** 10.3390/jimaging9020031

**Published:** 2023-01-30

**Authors:** Poul F. Høilund-Carlsen, Reza Piri, Oke Gerke, Michael Sturek, Thomas J. Werner, Mona-Elisabeth Revheim, Abass Alavi

**Affiliations:** 1Department of Nuclear Medicine, Odense University Hospital, 5000 Odense, Denmark; 2Research Unit of Clinical Physiology and Nuclear Medicine, Department of Clinical Research, University of Southern Denmark, 5230 Odense, Denmark; 3Department of Anatomy, Cell Biology, Physiology, Indiana University School of Medicine, Indianapolis, IN 46202, USA; 4Department of Radiology, Perelman School of Medicine, University of Pennsylvania, Philadelphia, PA 19104, USA; 5Division of Radiology and Nuclear Medicine, Oslo University Hospital, 0424 Oslo, Norway; 6Institute of Clinical Medicine, Faculty of Medicine, University of Oslo, 0315 Oslo, Norway

**Keywords:** atherosclerosis, disease progression, 18F-sodium fluoride, intervention, NaF, PET/CT

## Abstract

The method of 18F-sodium fluoride (NaF) positron emission tomography/computed tomography (PET/CT) of atherosclerosis was introduced 12 years ago. This approach is particularly interesting because it demonstrates microcalcification as an incipient sign of atherosclerosis before the development of arterial wall macrocalcification detectable by CT. However, this method has not yet found its place in the clinical routine. The more exact association between NaF uptake and future arterial calcification is not fully understood, and it remains unclear to what extent NaF-PET may replace or significantly improve clinical cardiovascular risk scoring. The first 10 years of publications in the field were characterized by heterogeneity at multiple levels, and it is not clear how the method may contribute to triage and management of patients with atherosclerosis, including monitoring effects of anti-atherosclerosis intervention. The present review summarizes findings from the recent 2¾ years including the ability of NaF-PET imaging to assess disease progress and evaluate response to treatment. Despite valuable new information, pertinent questions remain unanswered, not least due to a pronounced lack of standardization within the field and of well-designed long-term studies illuminating the natural history of atherosclerosis and effects of intervention.

## 1. Introduction

The method of 18F-sodium fluoride (NaF) positron emission tomography/computed tomography (PET/CT) imaging of atherosclerosis was introduced 12 years ago [1] and excels in showing arterial wall microcalcification long before the appearance of CT-detectable macrocalcification. It is vitally important that this early detection is at a time point when the atherosclerosis process may still be susceptible to intervention. In two reviews on NaF-PET in atherosclerosis we surveyed the literature until March 2020 [2,3]. Here, we summarize the literature since then to elucidate if remaining questions have been addressed, i.e., whether NaF-identified arterial microcalcification is a precursor of CT-detectable macrocalcification and whether it is possible to counteract the early phase of NaF uptake. The literature findings are presented below in six sections before being discussed with focus on novel observations of potential significance for the management of atherosclerosis.

## 2. Materials and Methods

Using the Patient, Intervention, Comparison, Outcome Study (PICOS) approach and systematic review principles [4], we searched PubMed/MEDLINE, Embase, Scopus, and Cochrane Library with the strings given in the Appendix A. Extracted were peer-reviewed articles in English from 1 April 2020 until 1 January 2023 with no restriction on comparator methods, outcome measures, or study design. Excluded were articles outside the scope of this review, editorials, letters, comments, conference proceedings, some methodology studies, reviews, and studies on amyloidosis or aneurysms. One experienced author (P.F.H.-C.) reviewed articles extracting information on number, sex, age, type of patients, tracer (NaF or NaF and 18F-fluorodeoxyglucose (FDG)), artery segment studied, purpose, quantification method, and main findings.

## 3. Results

Forty-five articles were included in this review [5,6,7,8,9,10,11,12,13,14,15,16,17,18,19,20,21,22,23,24,25,26,27,28,29,30,31,32,33,34,35,36,37,38,39,40,41,42,43,44,45,46,47,48,49] (Figure 1), to which were added selected articles on methodological and other aspects. Findings in the articles are summarized in the following sections and in Appendix A.

Disease mechanisms and targeting [5,6,7,8,9,10];Early detection and prevalence of NaF uptake in the heart and major arteries [11,12,13,14,15,16];NaF uptake in vulnerable, high risk, and ruptured plaque [17,18,19,20,21];Association between NaF uptake and risk factors [22,23,24,25,26,27,28];NaF uptake and disease progression or ‘prediction’ of events [29,30,31,32,33,34,35,36,37];Anti-atherosclerotic intervention evaluated by NaF-PET/CT [38,39,40,41,42,43,44,45,46,47,48,49].

Several studies dealt with one arterial bed only, i.e., coronary arteries [16,17,18,20,29,30,31,38,39,44], global heart [22,23,24,25,27,34], pulmonary arteries [11], left carotid artery [28], aorta [26,32,41,42,48,49], abdominal aorta [13,33,45], or several arterial beds [5,6,7,8,9,10,12,14,15,21,35,36,37,40,41,43,46,47]. Eight reports were experimental animal studies [5,6,32,42,43,45,48,49], and one was an image case report [44].

### 3.1. Disease Mechanism and Targeting

Zhuang et al. [5] started ApoE-/- rats on atherogenic diet at 13 weeks of age and found at 12, 27, and 46 weeks of age uptake of FDG, but not NaF, in the aortic arch and of NaF, but not FDG, in pulmonary arteries. FDG did not appear earlier than NaF. Pulmonary atherosclerotic lesions were fatty streaks with larger areas of inflammation (CD68 staining) and calcification (alizarin red staining) while in the aortic arch only a thickened intima was present but with very high expression of HIF-1α suggesting hypoxia. Unexpectedly, FDG uptake was not associated with inflammation, but hypoxia, whereas NaF uptake ‘correlated’ with microcalcification found in the pulmonary lesions only. The exact location of the tracer could not be determined due to insufficient PET resolution.

Nogales et al. [6] found in atherosclerotic transgenic Yucutan minipigs moderate co-localization of NaF uptake and CT-calcification in the heart, aorta, and iliac arteries and by autoradiography NaF accumulation in plaque calcifications in all examined arteries, correlating with both non-calcified plaque and thrombotic material. Some vascular segments had a NaF signal despite no CT-detectable calcification, which made the authors state decreased specificity and not increased sensitivity of NaF-PET compared to CT, despite that calcifications of <50 µm in size can be registered by NaF-PET, whereas size 200–500 µm is required for detection by CT [50].

Omarjee et al. [7] reported significant FDG and NaF uptake in diseased skin and in proximal aorta and femoral artery of pseudoxanthoma elasticum (PXE) patients. Aortic NaF uptake correlated, opposite to arterial FDG uptake, with pulse wave velocity (PWV). NaF uptake in skin regions correlated with calcium score. FDG and NaF uptake did not correlate in any arterial segment. Lastly, in dialysis patients with co-morbidities and many co-factors, Aaltonen et al. [8] observed a weak association between lumbar spine turnover mirrored by NaF activity and coronary calcification and between parathyroid hormone and coronary calcification. However, arterial wall calcification in PXE and dialysis patients is typically medical calcinosis, which may not follow the same NaF uptake mechanism as endothelial microcalcification.

Lastly, Wen et al. [9] demonstrated in 100 patients with multivessel disease a positive correlation between an average measure of NaF uptake proximally in all coronary arteries and NaF uptake in the arch and ascending aorta, and that coronary NaF activity is associated with peri-coronary adipose tissue density, by many is considered a player coronary atherosclerosis development due to exchange of pro-atherogenic mediators. Raynor et al. [10] followed another, also modern, trend by demonstrating higher NaF uptake in the coronary arteries and aorta in prostate cancer patients than in healthy controls, an elevation of about the same level as in patients with angina pectoris.

### 3.2. Early Detection and Prevalence of NaF Uptake in the Heart and Major Arteries

Zhang et al. [11] found that NaF uptake in the pulmonary arteries of angina pectoris patients, measured as standardized uptake values (SUVs) and target-to-background ratios (TBRs), was all significantly higher (11–36%) than in healthy controls. Quantifying NaF uptake in skin and in the carotids, aorta, iliac, femoral, and popliteal arteries of PXE patients, Gutierrez-Cardo et al. [12] noted highest uptake in the aorta and femoral arteries and in neck and axillae but no association between NaF deposition in other arteries and skin or the global Phenodex score indicating severity of PXE, whereas this score was significantly associated with vascular CT calcium score. Seraj et al. [13] analyzed NaF- and FDG-PET/CT scans in patients with rheumatoid arthritis (RA) and found significantly higher NaF TBRmean in the abdominal aorta of RA patients than healthy controls; the average CT calcium score was also higher, whereas FDG TBRmean scores were similar in the two groups.

In five patients with and five without peripheral arterial disease (PAD), Asadollahi et al. [14] found by NaF-PET/CT 30% higher total atherosclerotic burden in non-lower extremity arteries, i.e., coronaries, carotid artery, and aorta, in the PAD group due to higher carotid and thoracic aorta uptake than uptake in the coronaries and abdominal aorta. Bhattaru et al. [15] quantified atherosclerosis in upper and lower limb vessels of 68 healthy controls and 40 patients at-risk for cardiovascular (CV) disease and found 10% higher global cardiac NaF uptake in at-risk patients, but this global uptake was lower than in all other major arteries of both groups. In 114 male patients imaged with NaF PET/CT (for prostate cancer) and ^82^RB myocardial PET (for chest pain), Hayrapetian et al. [16] reported significantly higher coronary NaF uptake in patients with ischemic than normal myocardial perfusion. Among 41 patients with coronary angiography, coronaries with both obstructive and non-obstructive lesions had higher NaF uptake than coronaries without, but the correlation between CT-calcification and NaF uptake was poor.

### 3.3. NaF Uptake in Vulnerable, High Risk and Ruptured Plaque

Ashwathanarayana et al. [17] found that culprit plaques in 24 patients with ST-elevation myocardial infarction (MI) had 23–27% higher coronary NaF uptake than culprit plaques in 17 chronic stable angina patients. Majeed et al. [18] demonstrated by intra-coronary optical coherence tomography in 62 patients with acute coronary syndrome that coronary segments with elevated NaF uptake had higher lipid arc, higher prevalence of macrophages, lower plaque free wall, and higher total plaque burden and dense calcified plaque burden on CT-angiography than NaF negative segments, suggesting that coronary NaF uptake may further characterize coronary plaques and refine risk stratification in this category of patients. Mechtouff et al. [19] found clearly higher NaF uptake in culprit lesions than non-culprit plaques of carotid stenosis patients, a finding which was not associated with the morphological MRI criteria of vulnerability in plaque with stenosis >50%. Wurster et al. [20] observed that both calcified and non-calcified thin-cap fibroatheromas were predominantly located in coronary artery segments with NaF TBRmax >1.28, where fibroatheromas had significantly thinner caps. Finally, Kaczynski et al. [21] found significantly, but only slightly, higher NaF uptake in carotid culprit lesions, which, however, were significantly (*p* = 0.04) associated with MRI culprit lesion features (necrosis, intraplaque hemorrhage, ulceration, and calcification).

### 3.4. Association between NaF Uptake and Risk Factors

Rojulpote et al. [22] found in asymptomatic adults with no history of smoking, diabetes, or dyslipidemia, with normal blood pressure and without macroscopic CT calcification that global cardiac NaF uptake (cardiac SUVmean) is related to blood pressure. After adjusting for age and gender, diastolic and mean arterial blood pressures were independent ‘predictors’ of higher global cardiac NaF uptake. Patil et al. [23] observed in 69 healthy non-diabetic individuals a positive linear association of global cardiac NaF average SUVmean with total cholesterol/low-density lipoprotein ratio. Adjusted for age, gender, blood pressure, and multiple other factors, this ratio was independently associated with global cardiac average SUVmean. Gonuguntla et al. [24] studied global cardiac NaF uptake in 40 patients with CV risk and found that patients with higher CHADS2 and CHA2DS2-VASc scores, used to estimate risk of stroke in patients with atrial fibrillation, had a higher atherosclerotic NaF burden and could be at greater risk of CV events. Borja et al. [25] examined patients aged >40 years from the CAMONA trial with unknown risk divided into four ASCVD (Atherosclerotic Cardiovascular Disease) 10-year risk groups and observed increasing global cardiac NaF average SUVmean, from 0.67 for low risk, 0.70 for borderline risk, 0.72 for intermediate risk, and 0.78 for high risk, and that ASCVD risk score was significantly correlated to average SUVmean.

Paydary et al. [26] found in 89 healthy controls and 44 patients with chest pain a higher NaF uptake in three parts of the thoracic aorta in patients than in controls, and uptake in all three parts was correlated with age. In general, uptake in the entire thoracic aorta was a stronger predictor of Framingham Risk Score (FRS) than average SUVmax and average SUVmean and was a significant ‘predictor’ of an unfavorable CVD risk profile as compared with other values.

Borges-Rosa et al. [27] aimed to evaluate global cardiac (aortic valve included) microcalcification activity assessed by NaF imaging as a measure of unstable microcalcification burden in 34 high-risk patients with a previous CV event. The median global molecular calcification score (GMCS) of all studied patients was 320.9. Individuals with >5 CV risk factors (50%) had increased mean GMCS compared to patients with ≤5 risk factors (356.7 vs. 261.1), and this correlated positively with predicted fatal CV risk by SCORE. GMCS correlated positively with weight, body mass index (BMI), abdominal perimeter, thoracic fat volume, and epicardial adipose tissue but not with coronary calcium score or coronary artery wall NaF uptake.

Castro et al. [28] found in 128 patients with mixed CV risk that average NaF SUVmax in the left carotid artery was correlated with 10-year FRS, CHA2DS2-VASc score, and level of physical activity, was significantly higher in patients with increased risk of CV and thromboembolic events, and was significantly lower in patients with greater level of physical activity. Age, BMI, hypertension, and level of physical activity were independent associations of average SUVmax.

### 3.5. NaF Uptake and Disease Progression or ‘Prediction‘ of Events

Kwiecinski et al. [29] evaluated 293 patients with known CAD NaF TBRmax in proximal coronary arteries and found that fatal/non-fatal MI had after 42 months occurred exclusively in patients with abnormal NaF uptake at baseline: 20/203 vs. 0/90 in patients without abnormal coronary NaF uptake. Patients with increased NaF uptake had a >7-fold increase in MI independent of age, gender, CV risk, and other factors suggesting that abnormal coronary NaF uptake carries a significant risk of future MI.

Bellinge et al. [30] reported about 41 consecutive diabetes patients without a history of CAD but with a CT coronary calcification score (CCS) ≥ 10, who underwent a new CT scan two years after completion of a trial on effects of vitamin K-1 (see later). The proportion of “CCS progressors” was higher among NaF positive than NaF negative arteries at baseline (86.5% vs. 52.3%, *p* < 0.001), and NaF positive disease was an independent ‘predictor’ of subsequent CCS progression. 

Doris et al. [31] reported in 183 patients with multivessel disease and increased NaF uptake in at least one vessel that individuals with increased coronary NaF uptake had more rapid progression of calcification than had patients without uptake, that calcium score only increased in coronary segments with NaF uptake and not in segments without, that baseline coronary NaF TBRmax correlated with 1-year change in calcium deposition, and finally, that at the segmental level, baseline NaF activity was an independent predictor of calcium score at 12 months.

In ApoE-/- mice on a high-fat diet, examined at 12, 20, and 40 weeks of age, Hu et al. [32] observed gradually increased levels of lipoprotein, inflammatory cytokines, and calcifications factors in excised aorta of three mice at each time point and increased aortic NaF PET uptake with each weekly extension correlating with extent of calcification, the nature of which (microcalcification or macrocalcification), however, could not be determined. 

Fiz et al. [33] studied the infrarenal aorta of 71 patients imaged twice by NaF-PET/CT on average 15.5 months apart due to cancer and found multiple non-calcified (HU < 130) NaF hotspots with low, albeit higher, HU than non-calcified control regions (48 ± 8 vs. 37 ± 9, *p* < 0.01) and found that the HU of hotspots had increased (to 59) on the second scan. New calcifications appeared at the hot spot site in 41% of cases (Figure 2). Baseline atherosclerotic plaque (*n* = 375) NaF TBRmean was proportional to percent HU increase and increase in calcium score of existing plaques. Aortic calcium score increased, and whole-aorta NaF TBRmean at baseline correlated with the increase in calcium score from baseline to the second PET/CT.

Brodsky et al. [34] observed in 15 healthy males and eight female control subjects, who volunteered for a second NaF PET/CT after 2 years, only minimal changes in global cardiac NaF SUVmean and NaF SUVmax of +4% vs. −10%, respectively, and observed that CT-calcification measured in the same global cardiac ROIs decreased by 6%. The increase in SUVmean correlated with baseline BMI and systolic blood pressure in males but not in females.

Lillo et al. [35] found in 14 PXE patients imaged with NaF PET/CT two years apart an overall increase in CT calcification in major extracardial arteries, but this was due to an increase in two patients only; NaF SUVmax and SUVmean remained unchanged, while TBRmax and TBRmean decreased by 24% and 19%, respectively. However, as mentioned, it should be borne in mind that PXE is characterized by primarily medial calcification distinct from conventional atherosclerosis.

Reijrink et al. [36] investigated the major extracardial arteries in 10 type 2 diabetic patients without glucose lowering drugs and a severe CV history and found that baseline FDG TBRmean was strongly correlated with five-year follow-up NaF TBRmean, which correlated positively with change in calcified plaque score and with this score at both baseline and follow-up but not with change in calcified plaque score and PWV.

Piri et al. [37] found insignificant changes in the carotids and the aorta of healthy subjects and angina pectoris patients imaged 2 years apart but found higher NaF uptake in angina pectoris patients at both time points and found that baseline NaF could not predict change in subtle CT-calcification.

Kitagawa et al. [38] found no mean change in NaF uptake in 51 coronary lesions of 15 patients imaged twice 37–59 months apart. Baseline CT-based lesion feature (location, obstruction, plaque type, high-risk features) in individual patients did not correlate with change in NaF uptake, but 63% of NaF positive lesions at baseline were also positive at follow-up.

Kwiecinsky et al. [39] studied NaF activity in 154 coronary vascular grafts and found in mean 2.7 years after bypass surgery that all arterial graft and the majority (120/128) of venous grafts showed no NaF uptake. However, the bypassed coronary arteries had three times higher NaF activity than non-bypassed arteries and greater progression of 1-year CT-based calcium scores, an effect largely confined to native coronary plaques proximal to the graft anastomosis.

Fletcher et al. [40] analyzed scans of 461 patients with CVD to study associations between global NaF uptake in the epicardial coronary artery tree and the ascending part and arch of the aorta, respectively, and incidence of ischemic stroke and MI, which was 23 (5%) and 32 (7%), respectively, after in mean 6.1 years of follow-up. High thoracic NaF activity was strongly associated with ischemic stroke but not MI. Conversely, high coronary NaF activity was associated with MI but not ischemic stroke.

Finally, Dai et al. [41] found in a retrospective analysis of scans performed in 36 men with prostate cancer mainly weak correlations between proximal thoracic aortic NaF uptake and various cardiovascular risk factors, precluding any useful prediction of future vascular events.

### 3.6. Anti-Atherosclerotic Intervention Evaluated by NaF PET/CT

Hsu et al. [42] studied aortic root uptake in hyperlipidemic Apoe-/- mice; one group moved ad lib (*n* = 9), while the other had a progressive treadmill regimen for 9 weeks (*n* = 9). In vivo NaF µPET/µCT imaging demonstrated that while aortic calcification progressed similarly in both groups based on µCT, the fold change in NaF density was significantly less in the exercise group. Histomorphometric analysis of aortic root calcium deposits showed a lower mineral surface area of these in exercised than non-exercised mice. Exercise raised serum PTH levels twofold.

Florea et al. [43] reported NaF uptake in the left ventricle, aortic root, and arch of five wild type mice and 20 ApoE-/- mice split into four groups of five mice each. Wild type control mice were on a chow diet for 12 + 12 weeks; the four (Figure 3a–d,) ApoE-/- groups were all on a Western type diet for 12 weeks followed by (a) a standard or chow diet, (b) a Western type diet (advanced stage group), (c) the vitamin K MK-7 diet, or (d) a Warfarin diet for another 12 weeks. At the end of the study, the Warfarin group presented spotty calcifications by CT in the proximal aorta. All of the spots corresponded to dense mineralization on von Kossa staining. After the control, the MK-7 group had the lowest NaF uptake. The advanced and Warfarin groups presented the highest uptake in the aortic arch and left ventricle. The advanced stage group did not develop spotty calcifications, but NaF uptake was still observed, suggesting the presence of microcalcification.

In the third animal study in this section, Zhang et al. [45] compared two rabbit groups on a cholesterol-enriched diet, one of which also had atorvastatin (5 mg/kg/d). After 2 weeks of feeding, all underwent a dilated balloon operation of the abdominal aorta followed by a cholesterol-enriched diet for 16 weeks. NaF uptake increased significantly by 20% from the baseline until week 18 in the atherosclerosis group but only by <2% in the atorvastatin group. Total calcium density (assessed by von Kossa staining) was significantly increased (by 235%) in rabbits treated with atorvastatin compared with untreated rabbits, but the areas of microcalcification in plaque were significantly lower, and there were more microcalcification deposits in the areas with increased radioactive uptake of NaF.

Dietz et al. [44] published an image case story of a 64-year-old asymptomatic male with non-severe hypercholesterolemia and a history of limited smoking during teenage years. The patient was examined by cardiac NaF-PET/CT shown in Figure 3. The CT part showed a single mixed plaque near the ostium of the right coronary artery (a; RCA), a smaller plaque in the left main coronary artery (b; LM), and minor calcifications in the left anterior descending artery (c), while PET showed NaF focal uptake in the ostium of the RCA (d) and in the LM at baseline (e and f) and none in the left anterior descending. After 6 months on 10 mg rosuvastatin and 75 mg aspirin per day with advice on healthy diet and regular physical activity, low density lipoprotein cholesterol had decreased by 65%, and a second NaF-PET/CT revealed unchanged CT images and Agatston score (g, h, i) but a 37% decrease in focal NaF uptake in the RCA plaque (j) and a 40% decrease in the LM (k), without new hot spots in the LAD (i).

Bellinge et al. [46] examined 149 out of 154 type 2 diabetic patients with diabetes and CT coronary calcifications, who completed follow-up in a double-blind, placebo-controlled 2 × 2 factorial trial of 3 months duration and measured NaF TBRmax in proximal coronary arteries and ascending, arch, and descending parts of the thoracic aorta. In four treatments groups (placebo/placebo, vitamin-K1 [10 mg/day]/placebo, colchicine [0.5 mg/day]/placebo, and vitamin-K1 [10 mg/day]/colchicine [0.5 mg/day]), neither vitamin-K1 nor colchicine had a significant effect on coronary TBRmax compared with placebo. Later, according to a post hoc analysis of the same material [47] (applying an upper limit of normal for each examined arterial segment defined as the mean TBRmax + 2 standard deviations obtained in 10 patients of a non-treated diabetes cohort with zero coronary calcium), they found that vitamin K1 supplementation independently decreased the odds of developing new NaF PET positive lesions in the coronary arteries (odds ratio: 0.35) and aorta (odds ratio: 0.27) and in both aortic and coronary arteries (odd ratio: 0.28).

Jensen et al. [48] studied, in a randomized setup, the effects of semaglutide, a long-acting glucacon-like peptide-1 receptor agonist, on inflammation and calcification of the abdominal aorta of a non-diabetic atherosclerosis rat model. With PET/CT they found reduction in the uptake of the tracers ^64^Cu-DOTATATE and FDG mirroring activated macrophages and cellular metabolism, respectively, but no change NaF uptake, supporting the hypothesis that semaglutide inhibits atherosclerotic inflammation by means of decreased activated macrophage activity.

Finally, Bessueille et al. [49] investigated the effects of a tissue nonspecific alkaline phosphatase (TNAP) inhibitor (SBI-425) in an elaborated study of ApoE-deficient mice and found by NaF-μPET, μCT in vivo imaging, osteosense ex vivo imaging, and in vitro alizarin red staining that TNAP inhibition prevented calcification in mice aorta and in human smooth vascular muscle cells and, moreover, reduced blood cholesterol and triglyceride levels without impacting skeletal structures.

## 4. Discussion

The recent literature shows continued evidence for NaF PET/CT imaging in atherosclerosis. The discussion focuses on novel messages within the six subject areas.

### 4.1. Disease Mechanisms and Targeting

Selective concomitant uptake of FDG and NaF in the aortic arc and pulmonary arteries of obese ApoE-/- rats [5] is an observation not reported elsewhere. The study [6] in atherosclerotic minipigs could not tell for sure if NaF localizes in microcalcifications, but presence in arterial walls and thrombotic material was observed despite lack of CT detectable calcification. The dissociation between FDG and NaF uptake [7] in PXE patients suggested chronic ectopic calcification rather than acute inflammation, and the study [8] of dialysis patients revealed an interesting, but weak, association between bone metabolism and coronary calcification. The study [9] of patients with multivessel coronary disease demonstrated as expected a positive correlation between coronary and proximal aortic NaF uptake but also demonstrated that the former was associated with perivascular fat, as described by Borges-Rosa et al. [27] in patients with multiple CV risk factors. Finally, demonstration [10] of higher NaF uptake in the coronaries and aorta in patients with prostate cancer is one of series of publications [51,52] following the contemporary trend of studying molecular pathways shared by cancer and atherosclerosis.

### 4.2. Early Detection and Prevalence of NaF Uptake in the Heart and Major Arteries

NaF uptake in the pulmonary arteries, higher in angina pectoris patients than healthy subjects, is a new observation [11]. Dissociation between NaF uptake in skin and arteries of PXE patients is also a new observation [12], as is the aortic/pulmonary dissociation mentioned above between FDG and NaF uptake [7]. The lack of association between arterial NaF uptake and the Phenodex score [12] suggests similarities between skin and arterial calcification at the macrocalcification but not microcalcification level, but interpretation is difficult since CT-calcification develops slowly, not only in skin and arteries of PXE patients [35] but also in angina pectoris patients and healthy controls [34]. The higher uptake of NaF, but not of FDG, in the abdominal aorta of RA patients and greater average CT calcium volume than in healthy controls [13] and the higher FDG uptake in carotids and aorta of RA than osteoarthritis patients [53] are incomparable findings due to diverging quantification measures, i.e., TBRmax vs. blood background corrected SUVmax. Finally, the lower NaF uptake reported in the entire cardiac volume than observed in major extra-cardiac arteries of healthy and at-risk subjects may be due to “dilution”, as the small myocardial arteries contribute less [15], whereas the finding of higher coronary NaF uptake in patients with ischemic than normal myocardial perfusion appears more credible [16].

### 4.3. NaF Uptake in Vulnerable, High Risk, and Ruptured Plaque

It is noteworthy that patients with ST-elevation MI have significantly higher (~25%) NaF uptake in culprit plaques than have chronic stable angina patients [17] and that coronary segments with elevated NaF uptake have conventional characteristics of plaque building (higher lipid arc and prevalence of macrophages, lower plaque free wall, higher total plaque burden, and dense calcified plaque burden) when compared with NaF negative segments [18]. In carotid stenosis patients with and without symptoms and patients with a recent neurovascular event, NaF uptake was higher in culprit than non-culprit plaques [19,21], but in one study [19] NaF uptake was not associated with the morphological MRI criteria of vulnerability, whereas in the other [21] it was. In patients with multiple risk factors, both calcified and non-calcified coronary thin-cap fibroatheromas were predominantly located in segments with a TBR > 1.28, and plaques containing a lipid core were more frequent in segments with a TBR > 1.25 [20].

### 4.4. Association between NaF Uptake and Risk Factors

Even in healthy individuals, diastolic and mean arterial blood pressures and the ratio triglycerides/high density lipoprotein are independent ‘predictors’ of higher global cardiac NaF uptake [22,23]. Moreover, global cardiac NaF uptake is higher in patients with higher CHADS_2_ and CHA_2_DS_2_-VASc scores have also a higher global cardiac NaF burden [24], and patients aged >40 years present, when split into four ASCVD 10-year risk groups, with gradually increasing global cardiac NaF uptake, from 0.67 for low risk to 0.78 for high risk [25]. Further, patients with chest pain have higher NaF uptake in the thoracic aorta than healthy controls, and uptake in the entire thoracic aorta is a stronger predictor of FRS than average slice-base aortic SUVmax or SUVmean [26]. Patients with >5 CV risk factors have significantly higher cardiac “GMCS” than patients with ≤5 risk factors (356.7 vs. 261.1), and global cardiac NaF uptake correlates positively with fatal CV risk predicted by SCORE, besides with weight, BMI, abdominal perimeter, thoracic fat volume, and epicardial adipose tissue but not CCS or coronary artery wall NaF uptake [27]. Finally, average NaF SUVmax in the left carotid artery of patients with mixed CV risk correlates with 10-year FRS, CHA_2_DS_2_-VASc score, and level of physical activity and is significantly higher in patients with increased risk of CV and thromboembolic events and lower in patients with greater level of physical activity [28].

### 4.5. NaF Uptake and Disease Progression or ‘Prediction’ of Events

In the infrarenal aorta of cancer patients, the radiodensity of multiple non-calcified NaF hot spots increased from 48 to 59 HU in more than a year, while new calcifications appeared at hot spot sites in 41% of cases (Figure 2). Baseline average NaF uptake of atherosclerotic plaques (*n* = 375) was proportional to increase in calcium of existing plaques, and baseline whole-aorta NaF uptake correlated with 15-month increase in calcium score [33].

Four studies illustrated slowness and variability of the atherosclerotic process. One [34] reported in healthy controls after 2 years minimal changes in global cardiac NaF uptake, whereas CT-calcification in the same ROIs had decreased by 6%. Another [37] found in carotids and aorta of healthy controls and angina pectoris patients insignificant changes after 2 years, while in only two of 14 PXE patients there was a 2-year increase in CT-calcification in major extracardial arteries, whereas NaF SUVmax and SUVmean remained unchanged or decreased [35]. Finally, in a small sample [38] of patients with a total of 51 CT-defined coronary lesions, no change in an already low NaF uptake in the same lesions was observed after 37–59 months.

Five studies examined the relationship between arterial NaF uptake and future events or progression. In patients with known CAD, MI occurred only in those with abnormal baseline coronary NaF [29]; in diabetics, calcification progression was more common among NaF positive than NaF negative coronary arteries [30], while in patients with multivessel disease there was more rapid progression of CT-calcification among those with increased coronary NaF uptake and an increase in calcium score only in coronary segments with NaF uptake [31]. In major extracardial arteries of type 2 diabetic patients, there was a strong correlation of baseline FDG with NaF uptake measured at a 5-year follow-up [36], while in patients with prostate cancer there was weak association of aortic NaF uptake and several liquid atherosclerosis biomarkers, of little clinical impact [41].

In an analysis [40] of 461 patients with CVD, it was found after approximately one year that progression of thoracic aortic calcium volume correlated with baseline thoracic aortic NaF activity. Moreover, that in the 5% and 7% of patients, who had experienced an ischemic stroke or an MI, respectively, after six years of follow-up that high thoracic aortic NaF uptake was associated with stroke but not MI, and, conversely, that high coronary NaF activity was associated with MI but not ischemic stroke (Figure 4).

In the only study [39] to date examining NaF uptake in coronary vascular grafts, all 26 arterial grafts and 120/128 venous grafts were devoid of NaF activity almost three years after their insertion, but bypassed native coronary arteries had three times higher NaF uptake than non-bypassed arteries and greater progression in their calcium score, an effect largely confined to native coronary plaques proximal to the graft anastomosis.

### 4.6. Anti-Atherosclerotic Intervention Evaluated by NaF PET/CT

Three animal experiments illustrated facets of the same phenomenon, i.e., that intervention cannot stop arterial macrocalcification but may reduce NaF uptake due to more dense, but diminished, surface of progressed calcification. We postulate that this decrease in NaF uptake may also be due to inhibition of early phase NaF-avid microcalcification. One study [42] reported similar progression of CT-calcification in exercised and non-exercised Apoe-/- mice but less increase in NaF uptake in exercised mice, the latter apparently due to lower mineral surface area of aortic root calcification. Another [43] showed also in ApoE-/- mice that NaF PET/CT imaging is suitable for monitoring atherosclerosis development without or with only spotty CT-calcification. The third study [45] observed that NaF uptake in rabbit abdominal aorta had increased by 20% in the atherosclerosis group but hardly at all in the atorvastatin group. In contrast, total calcium density measured by CT had increased vastly. The human case of Dietz et al. [44] showed unchanged coronary plaques by CT after 6 months on a statin/aspirin and diet/exercise regimen but significantly decreased NaF uptake in these plaques and no new NaF hot spots. Collectively, these data suggest that combinations of statin, exercise, aspirin, and diet may in fact prevent early-phase arterial wall NaF uptake (Figure 3). The two articles on vitamin-K1 treatment in diabetic patients showed diverging results: initially no change in coronary or aortic NaF uptake [46], but later, using a newly defined, potentially questionable [54], upper limit of normal for TBRmax, the vitamin-K1 groups had decreased odds of developing new NaF positive lesions [47].

Finally, recent animal studies showed interesting effects of intervention. One [48] found in a non-diabetic rat model that semaglutide, a glucacon-like peptide-1 receptor agonist, inhibits inflammation but not microcalcification in the abdominal aorta, while the other [49] observed in ApoE-deficient mice on a high fat diet that blocking of TNAP, a common enzyme necessary for normal bone and teeth formation but potentially also responsible for atherosclerotic plaque calcification, prevented calcification in mice aorta and in human smooth vascular muscle cells and, moreover, reduced blood cholesterol and triglyceride levels without impacting skeletal structures.

### 4.7. Methodology

Standardization is severely lacking. Guidelines exist for PET in cancer and NaF PET/CT bone scans [55,56,57], for quality assessment of scans and harmonization of PET/CT systems [57,58], but not for NaF-PET imaging in atherosclerosis. For quantification, most authors used SUVmax with or without subtraction of blood pool activity or TBRmax, i.e., SUVmax divided by blood activity, disregarding caveats in focusing on single voxels or variations in blood pool NaF activity [59]. Moreover, TBR is sensitive to acquisition timing and less suited for assessment of changes over time [60,61]. However, this does not preclude repeat imaging if the injection-scan time interval is noted and used again at re-imaging. Finally, authors often forgot to report use of same or different PET/CT scanners at repeat scanning, and information about relevant medication is often sparse.

New procedures to correct for cardiac movements are commendable efforts [62,63,64] but relate exclusively to NaF uptake in the proximal part of large epicardial coronary arteries, representing but a fraction of the entire cardiac atherosclerotic burden, which may be clinically more relevant [65,66], although it still remains to be fully demonstrated that all cardiac NaF uptake originates from the artery wall and not from other structures [67]. Noteworthy are initiatives that may increase the clinical usefulness of NaF-PET/CT. One such approach is fused assessment of peak systolic wall shear stress measured by 4D magnetic resonance imaging and NaF-PET to characterize disturbed aortic vascular function [68]. Another is the combination of measurement of average uptake in vessel whole sections, like major parts of or whole aorta [37,69,70,71], or in the entire heart [23,24,25,26,27,72,73]. The latter approach compensates for cardiac movements and shortcomings in analyzing only part of cardiac atherosclerosis burden. A particular challenge when measuring NaF uptake in arteries is to avoid spill-over from nearby bony structures, such as the vertebrae when examining uptake in the aorta. Some authors describe observing a safety distance of a certain size [33]; thus, we excluded high uptake (>2.5 × SD) in regions of the aorta closer than 1 cm to the vertebral bone body [71], whereas others did not mention the problem-which obviously call for a proper solution and standardization. These many efforts call for a second initiative, i.e., systems that can facilitate, improve, and speed up manual or semi-automated, time-consuming, and observer-dependent segmentation. An efficient answer to this appears to be artificial intelligence-based processing, which, besides minimizing observer dependence and being much faster (Figure 5), is constantly being improved, while at the same time allowing consideration also of multiple other relevant data [72,73,74,75,76].

### 4.8. Limitations

Limitations of animal studies like insufficient spatial PET resolution [5,6,32,42,43,45] hamper extrapolation to human conditions. A significant advantage is the use of human-sized animal models that enable NaF-PET/CT imaging of large vessel calcification and microvascular function at multiple time points during atherosclerosis pathogenesis and during prevention and regression studies [6,75]. Human studies had other limitations. Many were parts of trials with other primary aims, post hoc analyses, or studies designed to investigate other diseases rather than the role of NaF-PET in atherosclerosis, and longitudinal studies with an adequate number of repeat scans [76] were not reported. The word ‘prediction’ was used over and over when a statistical association was demonstrated by group comparison [22,23,29,30,36], but no reported association was close enough to allow prediction of the course of individual patients with even a reasonably high degree of certainty, which is why we have sometimes written ‘prediction’ surrounded by apostrophes.

### 4.9. Summing Up

Valuable new information supports the interpretation that arterial NaF uptake is a precursor of CT-calcification. However, it is not clear when in the time course declining NaF uptake reflects a decrease in atherosclerosis development or rather limited adsorption secondary to late-phase consolidation of CT-detectable calcification. Very early atherosclerosis development, i.e., prevention studies, is needed to determine whether the initial NaF uptake steps can be attenuated and thereby decrease atherosclerosis. There is a pronounced lack of standardization within the field and of well-designed long-term studies to illuminate the natural history of atherosclerosis and effects of intervention in humans. NaF-PET is still a research tool, but the use of artificial intelligence procedures bodes well for potential implementation in the clinical routine.

## Figures and Tables

**Figure 1 jimaging-09-00031-f001:**
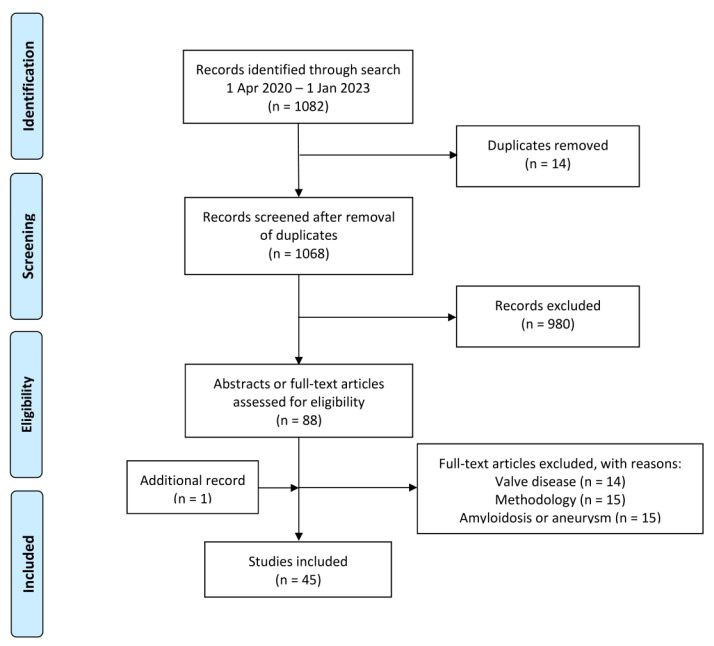
PRISMA Flow Diagram.

**Figure 2 jimaging-09-00031-f002:**
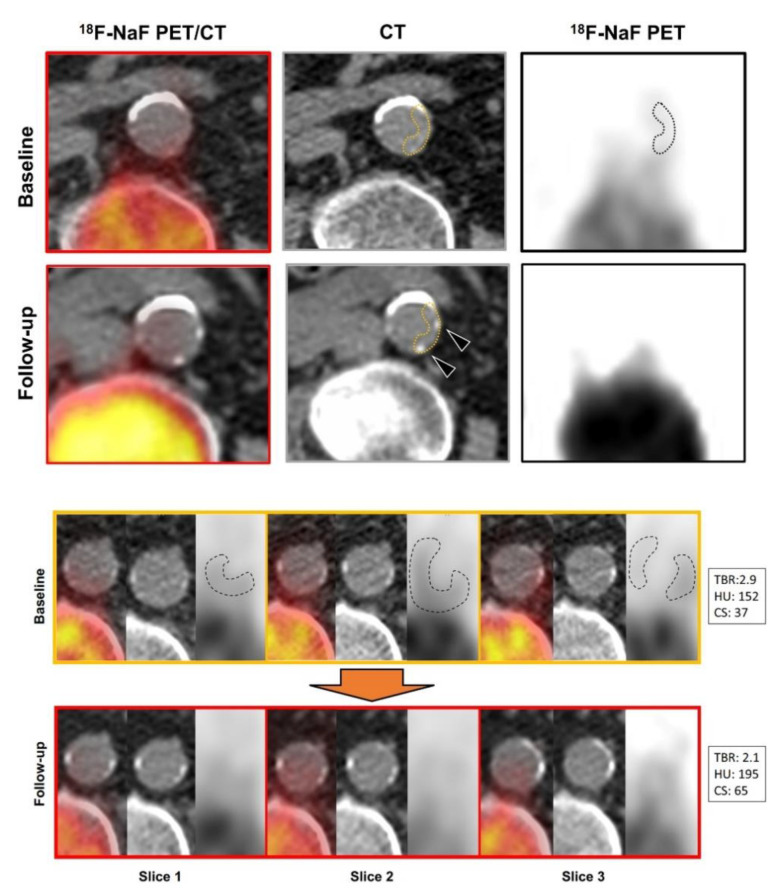
Appearance of Calcifications in Areas of Increased NaF Uptake and Evolution of “Hot” Plaques. Top panel: Fused and PET images show increased NaF uptake in the posterior and left lateral aortic wall (top row, left and right images, dashed outlines). At follow-up, two point-shaped calcifications have appeared (bottom row, left and center images, see arrowheads). Bottom panel: The upper series (orange box) shows three consecutive NaF-PET/CT and CT slices with mildly calcified plaques with co-localized increase in tracer uptake (dashed outlines). Follow-up images of the same slices (red box) show an increase in the lesions’ density and size, the extent of which is reflected by the increase in average HU and calcium score. TBR values were average SUV values normalized for inferior vena cava blood pool activity. Spill-over from nearby vertebrae were sought corrected for by means of brush tool editing observing a safety distance of 1 cm. Reprinted with permission from Fiz et al. [33], 2021, Springer Nature.

**Figure 3 jimaging-09-00031-f003:**
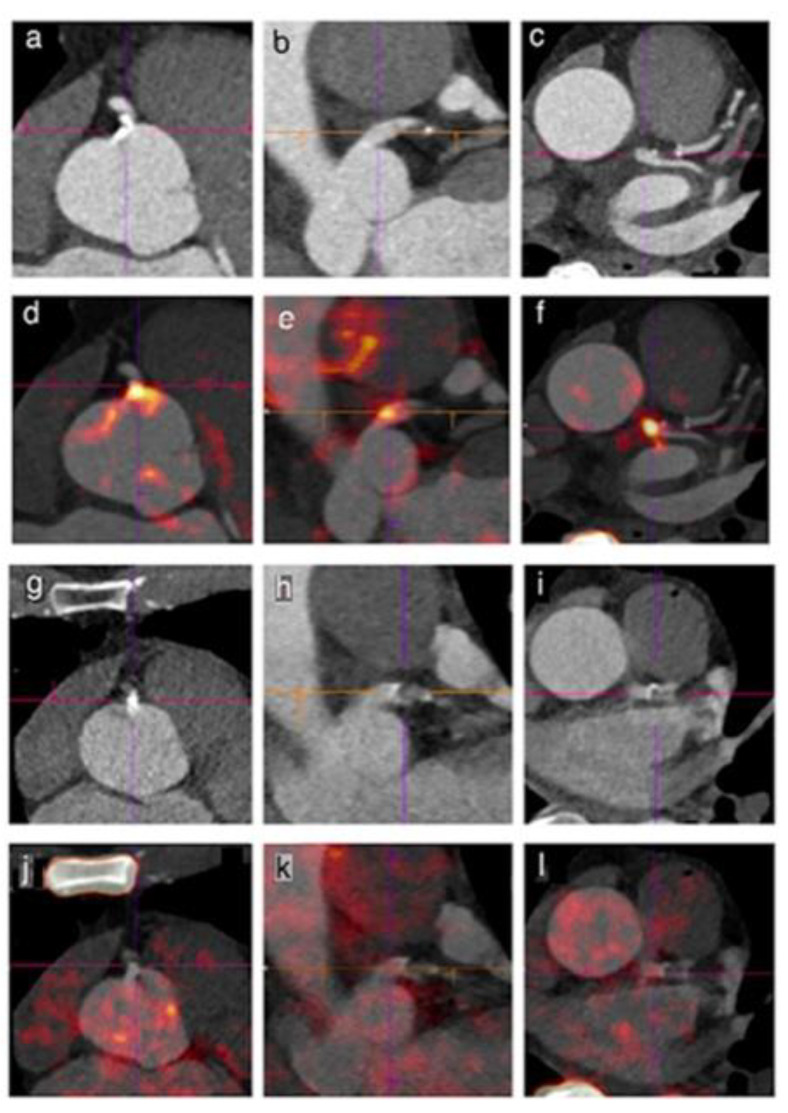
Reduced 18F-Sodium Fluoride Activity in Coronary Plaques after Statin Therapy. CT: single mixed plaque extending to the ostium and first segment of right coronary artery (RCA), smaller one in left main coronary artery (LM) and minor calcifications in left anterior descending artery (LAD); Agatston scores: ostium 198; RCA 169; LM 22; LAD 7 (panels (**a**–**c**)). NaF-PET/CT: focal NaF uptake in ostium of RCA (panel **d**; SUVmax1.9) and in LM (panels (**e** and **f**); SUVmax 2.5). Six months later following a regimen of 10 mg rosuvastatine and 75 mg aspirin per day with advising of diet and regular physical activity, CT was unchanged with Agatston scores: ostium 195; RCA 163; LM 23, LAD 9, while there was a decrease in focal NaF uptake in the coronary plaques (panels (**g**–**l**); respective SUVmax 1.2 and 1.5), without new hotspots. Reprinted with permission from Dietz et al. [44]). 2021, Oxford University Press.

**Figure 4 jimaging-09-00031-f004:**
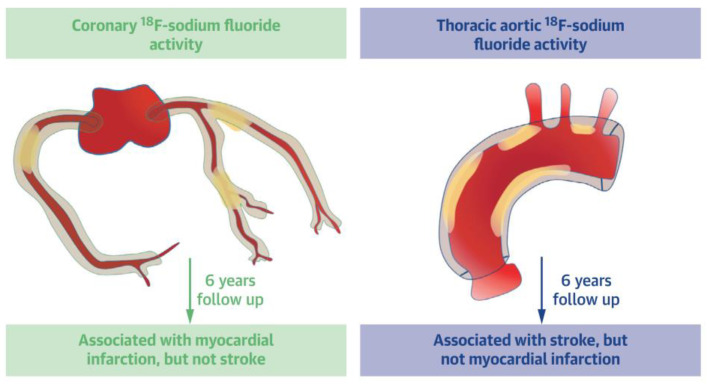
Schematic representation of measurement NaF uptake and calculation of coronary microcalcification score (**left**) and arch of the aorta NaF activity (**right**). Reproduced from Fletcher, A.J et al. [40], an open access article under the CC BY license.

**Figure 5 jimaging-09-00031-f005:**
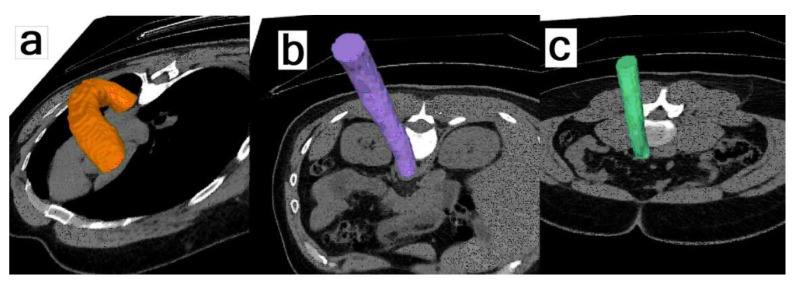
Artificial Intelligence-Based Segmentation of the Aorta. Convolutional neural networks trained on manually annotated non-contrast CT images allows reconstruction of the arch of aorta (**a**), thoracic aorta (**b**), and abdominal aorta (**c**). After identifying the edge of the aorta, segmentation of the wall (not shown) is made by including all voxels lying within 3 mm of the edge of the aorta segmentation on the inside and within 2 mm of the edge on the outside, yielding a 5 mm thick wall following the edge of the aorta segmentation output from the segmentation tool. This yielded a 5 mm thick wall following the edge of the aorta. Values obtained with this approach for average NaF SUVmean uptake in the walls of three sections were very virtually identical to values obtained by manual segmentation. Reprinted with permission from Piri et al. [71]). 2021, Springer Nature.

## Data Availability

Not applicable.

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
