# Peer review of "NaF-PET Imaging of Atherosclerosis Burden"

_2313-433X, 2023, doi:10.3390/jimaging9020031_

Round 1

Reviewer 1 Report

The paper entitled “NaF-PET imaging of atherosclerosis burden” is a systematic review well written, with purposes established in the introduction of the manuscript:

1.      whether NaF-identified arterial microcalcification is a precursor of CT-detectable macrocalcification

2.      whether it is possible to counteract the early phase of NaF uptake

According to the literature it is known that sodium fluoride is a marker of microcalcification and its uptake, in the arterial wall, is related with the cardiovascular risk of events. In 2014, in The Lancet, a prospective clinical trial that involved patients with myocardial infarct, angina pectoris and symptomatic carotid disease, concluded about the relation of this tracer uptake with culprit plaques in myocardial infarct patients, culprit carotid plaques and vulnerable plaques in patients with angina. Since then numerous studies were published on the subject and regarding micro and macrocalcification it seems that in the arterial wall the first precedes the second, though more longitudinal studies are necessary to show how microcalcification activity could be translated into macrocalcification and how its modification could influence cardiovascular risk. It is also clear that atherosclerotic burden is not the same as vulnerable plaque burden and cardiovascular risk of events. Stable plaques tend to be calcified whether the vulnerable plaques have microcalcification and inflammation among other characteristics. In fact, statins reduce 18F-FDG uptake (inflammation) in the arterial wall and they also seem to increase calcification.

In this paper these concepts are not well explained and should be clarified.

Examples:

 “Valuable new information supports the interpretation that arterial NaF uptake is a precursor of CT-calcification. However, it is not clear whether declining NaF uptake reflects a decrease in atherosclerosis burden or is simply the limited adsorption due to consolidated CT-detectable calcification

Very early atherosclerosis development prevention studies are needed to determine whether the initial NaF uptake steps can be attenuated and thereby decrease atherosclerosis.”

A decrease in atherosclerotic burden or a decrease in atherosclerosis, don’t seem to be adequate expressions; vulnerable plaque burden and plaque stabilization should be used instead.

The methodology is well explained and according to the authors, 36 articles, published between 2020 and 2022, were selected. They were summarized in six sections and discussed in three; the six sections considered in the results were “Disease Mechanism and Targeting”, “Early Detection and Prevalence of NaF Uptake in the Heart and Major Arteries”, “NaF Uptake in Vulnerable, High Risk and Ruptured Plaque”, “Association Between NaF Uptake and Risk Factors”, “NaF Uptake and Disease Progression or ‘Prediction‘of Events” and “Anti-Atherosclerotic Intervention Evaluated by NaF PET/CT”. In the discussion the three sections were: “Early Detection and Prevalence of NaF Uptake in the Heart and Major Arteries”, “NaF Uptake and Disease Progression or ‘Prediction‘ of Events” , “Anti-Atherosclerotic Intervention Evaluated by NaF PET/CT” and “Methodology”.

It is difficult to understand the six sections in the results. It would be clear if the sections were the same in the results and discussion sections.

In the discussion should be avoided to repeat information already given in the results.

Some methodologic limitations are recognized by the authors in the reviewed papers, namely, the lack of standardization in the use of sodium fluoride in the evaluation of atherosclerosis. In fact this is still a research method far from a broad clinical application and so this limitatipn is relative.

Tables (in the supplementary material) and figures are very enlightening.

Reviewer 2 Report

Review of 18F NaF covering advancements from April 2020 to April 2022. Review assesses new evidence linking 18F NaF with atherosclerotic burden with a focus where the technique is regarding transition to clinical use.  The review has a clear methodology and search strategy and results and discussion broken down into easy-to-read sub-sections.

Overall, the review - authored by recognised experts in the area - is well written, thoughtful, and will be of interest to imaging experts; both those wanting to understand potential uses for 18NaF imaging and those more experienced with it wanting an up-to-date review.

While the search strategy is robust - it is now 6 months out of date - and there have been significant steps forward in this time.  While recognising there is some additional work, this reviewer believes it will be a more cutting-edge review to capture.  Specifically, in the discussion the authors convincingly outline the high heterogeneity of method used to assess NaF uptake and that non-standardization is a problem.  Given this, and the emergence of artificial intelligence (Piri et al doi: 10.1111/cpf.12793, Singh et al doi: 10.1007/s12350-022-03010-8), which has the potential to bring NaF PET much closer to clinical use,  a designated sub-section on methods for measuring NaF would be appropriate, highly relevant and interesting as well as helping to contextualise the findings discussed throughout.

Based on a repeated the authors search using the criteria in the supplement on 13/12/22 there were 12 studies that fit into the authors designated categories, many of which are important steps forward. Specific publications are highlighted below:

Disease mechanism and targeting.

Minderhoud et al doi: 10.1093/ehjci/jeac090

Early detection and prevalence of NaF uptake in the Heart and Major Arteries

Dai et al doi: 10.3390/ijms232113056

Kwiecinski et al doi: 10.1016/j.jcmg.2021.11.030

Uptake in Vulnerable and High Risk and Ruptured Plaque

Kaczynski et al doi: 10.1148/radiol.212283

Wurster et al doi: 10.1093/ehjci/jeab276

Association between NaF and risk factors

Kitagawa et al doi: 10.1007/s12350-022-02975-w

Wen et al doi: 10.1007/s12350-022-02958-x

Naf Uptake in Disease Progression or Prediction of Events

Fletcher et al doi: 10.1016/j.jcmg.2021.12.013

Anti-atherosclerotic intervention evaluated by NaF

Jensen et al doi: 10.1016/j.atherosclerosis.2022.03.032

Bessueille et al doi: 10.1016/j.trsl.2022.06.010

Minor point

If possible, for the figures 2 and 3, SUV/TBR color scale would be helpful in contextualising the results presented.

Round 2

Reviewer 1 Report

The review is more understandable and the text is properly structured. The concepts were clarified.